# Peer review of "A Colorimetric/Ratiometric Fluorescent Probe Based on Aggregation-Induced Emission Effect for Detecting Hypochlorous Acid in Real Samples and Bioimaging Applications"

_foods, 2025, doi:10.3390/foods14142491_

Round 1
Reviewer 1 Report
Comments and Suggestions for Authors
The current manuscript discusses a pyrene-based AIE-active sensor probe for the detection of hypochlorous acid. The work presents interesting findings centered on a dual-emissive sensor probe. However, the manuscript requires more robust data presentation and additional experimental evidence before it can be considered for publication. Therefore, I recommend a major revision of the current manuscript. Please see my detailed comments below:
- The authors should discuss the advantages of a dual-emission probe over a single-emission sensor in the Introduction section. This comparison will help readers understand the significance of the proposed system.
- The manuscript shows a high similarity index, particularly in the Experimental section, figure captions, and other parts. The authors must work on rewriting these sections to reduce the similarity and improve originality.
- The Introduction should include a discussion on how the current probe addresses the limitations of previously reported sensors, highlighting its unique features and performance advantages.
- Recent developments in pyrene-based dual-emissive/colorimetric sensors should be cited to provide a broader context for the work. The authors may consider referencing the following articles:
a) https://doi.org/10.1021/acsami.1c04744
b) https://doi.org/10.1039/D0DT00837K - Although the probe is described as AIE-active, no experimental evidence supporting AIE activity is presented. This data is crucial and must be included in the manuscript.
- Figures 1b and 1c should be combined into a single plot showing concentration-dependent fluorescence titration spectra, along with corresponding visual changes under a UV lamp at various HOCl concentrations. A similar approach should be used for UV–Vis spectra. Additionally, a CIE chromaticity plot should be provided.
- The authors should include a supplementary data file. The graphs currently shown in Figures 2a and 2b should be moved to the supplementary material, and the data should instead be summarized in the main manuscript using bar plots with error bars representing triplicate measurements.
- The effect of pH on the sensing performance should be discussed. Specifically, the authors should explain why the probe works best at pH 8, and what changes occur at lower or higher pH values.
- To confirm the sensing mechanism, NMR and mass spectra of the oxidized probe (in the presence of HOCl) should be provided.
- The figure numbering in the manuscript should be consistent. Instances of both “2a” and “4A” are used—this inconsistency must be corrected.
Author Response
The current manuscript discusses a pyrene-based AIE-active sensor probe for the detection of hypochlorous acid. The work presents interesting findings centered on a dual-emissive sensor probe. However, the manuscript requires more robust data presentation and additional experimental evidence before it can be considered for publication. Therefore, I recommend a major revision of the current manuscript. Please see my detailed comments below:
Comments 1: The authors should discuss the advantages of a dual-emission probe over a single-emission sensor in the Introduction section. This comparison will help readers understand the significance of the proposed system.
Response 1: Thank you for your valuable suggestions. We agree that discussing the advantages of dual-emission probes over single-emission probes in the Introduction will be beneficial to readers. We have accordingly revised the introduction section to highlight these advantages, including improved detection accuracy, reduced environmental interference, and enhanced reliability (Lines 70–73, Page 2). These features emphasize the significant importance of our dual-emission probes in advancing the detection of HClO.
Comments 2: The manuscript shows a high similarity index, particularly in the Experimental section, figure captions, and other parts. The authors must work on rewriting these sections to reduce the similarity and improve originality.
Response 2: We thank the reviewer for the comprehensive assessment of our manuscript and express our gratitude for the concerns raised regarding the similarity index of the experimental section (Lines 84–170, Pages 2–4), the captions of the images (for Scheme 1, Line 191, Page 5; for Figure 1, Lines 195–199, Pages 6–7; for Figure 2, Lines 231–235, Page 8; for Figure 3, Lines 274–275, Page 9; for Figure 4, Lines 307–309, Page 11; for Figure 5, Lines 339–343, Page 12; for Figure 6, Lines 362–364, Page 13), and other parts. We have carefully reviewed the proposed revisions and taken significant steps to rewrite and reorganize these chapters to enhance originality and reduce similarity.
Comments 3: The Introduction should include a discussion on how the current probe addresses the limitations of previously reported sensors, highlighting its unique features and performance advantages.
Response 3: We sincerely appreciate the reviewer’s comment regarding the need to more explicitly highlight how the current probe, NYV, addresses the limitations of previously reported probes. We provide a detailed revision to the Introduction section, incorporating a focused discussion on the unique features and performance advantages of NYV compared to existing probes (Lines 63–73, Page 2).
Comments 4: Recent developments in pyrene-based dual-emissive/colorimetric sensors should be cited to provide a broader context for the work. The authors may consider referencing the following articles:
a) https://doi.org/10.1021/acsami.1c04744
b) https://doi.org/10.1039/D0DT00837K
Response 4: We sincerely appreciate the reviewer’s suggestion regarding the inclusion of recent developments in pyrene-based dual-emissive/colorimetric probes. These references will indeed provide a broader context for our work and highlight the advancements in this field (for Reference 5, Lines 399–400, Page 14; Reference 32, Lines 467–469, Page 15).
Comments 5: Although the probe is described as AIE-active, no experimental evidence supporting AIE activity is presented. This data is crucial and must be included in the manuscript.
Response 5: Thanks for your helpful comments and advice. We appreciate the opportunity to improve our manuscript. In response to your concern regarding the lack of experimental evidence supporting the AIE activity of probe NYV, we have conducted additional experiments and prepared the necessary data to include in the manuscript (Lines 249–260, Pages 8–9, Figure S3).
Comments 6: Figures 1b and 1c should be combined into a single plot showing concentration-dependent fluorescence titration spectra, along with corresponding visual changes under a UV lamp at various HOCl concentrations. A similar approach should be used for UV–Vis spectra. Additionally, a CIE chromaticity plot should be provided.
Response 6: Thanks for your helpful comments and advice. In response to your recommendation to combine Figures 1b and 1c into a single plot, we carefully considered the advantages of integrating the concentration-dependent fluorescence titration spectra with visual changes under UV light. While such a combined plot could enhance the correlation between spectral data and visual observations, we opted to retain the current format for the following reasons:
Separating the visual changes (Figure 1b) from the quantitative fluorescence titration spectra (Figure 1c) allows readers to distinctly appreciate the intuitive colorimetric response (under UV light) and the precise spectral shifts (measured by fluorometry). This avoids the potential overcrowding of information in a single figure.
The fluorescence titration spectra in Figure 1c provide a detailed, instrument-based validation of the probe’s sensitivity, which is critical for reproducibility and quantitative analysis. Combining it with visual snapshots might dilute the emphasis on these quantitative results.
The insets in Figures 1a and 1b already highlight the visual changes in solution color and fluorescence, serving as a direct counterpart to the spectral data in Figure 1c. This ensures readers can easily correlate the two aspects without merging the figures.
To further address the reviewer’s concern, we have now included a CIE 1931 chromaticity diagram (Figure 1f) to objectively quantify the colorimetric changes during HClO titration (Line 194, Page 6). This addition bridges the gap between visual observations and instrumental measurements, providing a standardized evaluation of color shifts that aligns with the reviewer’s request for a more rigorous analysis.
We hope this explanation clarifies our rationale for the current figure organization while demonstrating our commitment to incorporating the reviewer’s feedback. Should the reviewer still prefer a combined figure, we would be happy to explore this in a revised submission.
Comments 7: The authors should include a supplementary data file. The graphs currently shown in Figures 2a and 2b should be moved to the supplementary material, and the data should instead be summarized in the main manuscript using bar plots with error bars representing triplicate measurements.
Response 7: We appreciate the reviewer’s constructive suggestion regarding the presentation of selectivity and anti-interference data. As noted in our original submission, the detailed fluorescence spectra for probe NYV in the presence of various interfering species are provided in Figure S2 of the Supplementary Material.
Comments 8: The effect of pH on the sensing performance should be discussed. Specifically, the authors should explain why the probe works best at pH 8, and what changes occur at lower or higher pH values.
Response 8: Thanks for your helpful comments and advice. Probably, ionization of the fluorescent molecules, resulting from changes in pH, alters the molecular orbital of the excitable electrons, and this could cause the intensity and/or position of the emission maximum to change (Marine Chemistry, 1982, 11, 395−401; Water Research, 2002, 36, 2571−2581). At lower pH values, the increased concentration of H+ ions may lead to protonation of the probe's functional groups, reducing their reactivity towards HClO. Conversely, at higher pH values, the functional groups in the probe may also become deprotonated.
Comments 9: To confirm the sensing mechanism, NMR and mass spectra of the oxidized probe (in the presence of HOCl) should be provided.
Response 9: We are incredibly grateful to the reviewers for their insightful and constructive suggestions, providing NMR and mass spectrometry of the oxidation probe (in the presence of HOCl) as evidence to further confirm the sensing mechanism. Although direct nuclear magnetic resonance and mass spectrometry data for the probe's oxidation form have not been obtained in this paper yet, we carefully analyzed the spectral data obtained from our study to infer the sensing mechanism and confirm the interaction between the probe and HClO.
Based on the observed spectral changes, we propose that the interaction between probe NYV and HClO leads to the oxidation of the carbon-carbon double bond within the probe molecule, which is highly sensitive to the oxidation of HClO. Similar literature reports indicate that HClO oxidizes the carbon-carbon double bond (Chemical Science 2018, 9, 8207–8212; Spectrochimica Acta Part A: Molecular and Biomolecular Spectroscopy 2024, 321, 124754; Sensors and Actuators B: Chemical 2018, 276, 8–12). Oxidation events alter the electronic structure of the probe, resulting in significant spectral shifts and intensity changes. Specifically, the reduction in absorption at 500nm and the transformation of fluorescence emission from orange-yellow to blue (as shown in Figures 1a and 1b) indicate a structural shift consistent with the expected oxidation mechanism. The color change from yellow to colorless in solution (inset of Figure 1a) and the fluorescence change from orange-yellow to blue (inset of Figure 1b) are particularly striking, providing visual confirmation of the significant structural changes occurring in the probe. These observations further strengthen our hypotheses regarding the oxidation mechanism and the interaction between probe NYV and HClO.
We recognize that the proposed oxidation mechanism is consistent with previous reports on the detection of fluorescent probes by HClO, and the oxidation event leads to significant spectral changes. Furthermore, the response of our probe to various interfering substances can be disregarded (Figure S3), indicating that our probe exhibits high sensitivity and selectivity, which strongly supports the specificity of the interaction between probe NYV and HClO.
We fully agree with the reviewer that the fluorescence response mechanism is necessary and relevant. For this reason, the relevant content is subject to further research in detail, and the results will be published in a separate article.
Comments 10: The figure numbering in the manuscript should be consistent. Instances of both “2a” and “4A” are used—this inconsistency must be corrected.
Response 10: We sincerely thank the reviewers for their meticulous attention to the details. We have thoroughly reviewed the manuscript and standardized all graph and subgraph labels to ensure consistency (for Figure 4, Line 306, Page 10; for Figure 5, Line 338, Page 12; for Figure 6, Line 361, Page 13).
Reviewer 2 Report
Comments and Suggestions for Authors
This manuscript is on the presentation of a colorimetric- fluorescent chemosensor (NYV) with good sensitivity and selectivity toward HClO. As fluorescent chemosensor, the sensibility is reached though a ratiometric approach. The chemosensor is built from coumarine and pyrene fragments. In overall, the manuscript is of interest but requires revision.
1) Although the ratiometric sensing is proved, the emission of the chemosensor NYV seems to be anomalous. From Figure 1, the bandwidth of the absorption band ranges from 450 nm to 525 nm. Surprisingly, the emission is composed of two bands, one high energy emission band from 425 nm to 525 nm and a low energy emission band from 550 – 650. The high energy emission band fully overlaps with the absorption band. What is the origin of such anomalous emission spectra? Please discuss this issue using a Jablosnky diagram. This discussion is important to provide a clear understanding of the ratiometric properties of the hybrid NYV molecule
2) Please add to SI the absorption and emission spectra of the coumarine and pyrene fragments and discuss the effect upon the bonding of the fragments.
3) Oines 162. It is mentioned that pyrene provides the property of AIE and substantial Stokes shift. Please discuss more clearly these two properties for the hybrid molecule NYV. In other words, quantify the AIE property and mention what is the Stokes Shift of NYV. Please consider the dual emission band of this molecule (Figure 1)
4) Lines 238. What does “fluorescent illumination” mean? Is it referred to UV illumination?
5) Line 210. Is it correct the “Despite interference”? Plots in Figure 2 don’t show any interference
6) Figure 3. The upper row Light stand for White light illumination?.
7) What is the site within the NYV molecule which sense HClO?
Author Response
This manuscript is on the presentation of a colorimetric- fluorescent chemosensor (NYV) with good sensitivity and selectivity toward HClO. As fluorescent chemosensor, the sensibility is reached though a ratiometric approach. The chemosensor is built from coumarine and pyrene fragments. In overall, the manuscript is of interest but requires revision.
Comments 1: Although the ratiometric sensing is proved, the emission of the chemosensor NYV seems to be anomalous. From Figure 1, the bandwidth of the absorption band ranges from 450 nm to 525 nm. Surprisingly, the emission is composed of two bands, one high energy emission band from 425 nm to 525 nm and a low energy emission band from 550 – 650. The high energy emission band fully overlaps with the absorption band. What is the origin of such anomalous emission spectra? Please discuss this issue using a Jablosnky diagram. This discussion is important to provide a clear understanding of the ratiometric properties of the hybrid NYV molecule
Response 1: Thanks for your helpful comments and advice. Thanks for the suggestion. Typically, the wavelength of fluorescence emission is longer than the absorption wavelength because the excited molecule loses some of its energy through vibrational relaxation. The emission spectrum of the probe is abnormal because the emission does not result from the energy transition of S1 → S0. Still, in the direct transition of higher energy levels (S2→S0), the excited state is rapidly equilibrated. The rapid internal conversion between S2 and S1 is suppressed, allowing for the phenomenon of complete overlap between the high-energy emission band and the absorption band to occur (Figure R1).
Figure R1 Jablonsky diagram of probe NYV (S0: ground state; S1, S2: singlet excited states; T1, T2: triplet excited states).
Comments 2: Please add to SI the absorption and emission spectra of the coumarine and pyrene fragments and discuss the effect upon the bonding of the fragments.
Response 2: We sincerely appreciate the reviewer’s insightful suggestion to include the absorption and emission spectra of the coumarin and pyrene fragments and discuss the effect of their bonding on the probe’s properties. The coumarin fragment exhibits absorption at 437 nm and emission at 478 nm, while the pyrene fragment shows absorption at 365 nm and emission at 428 nm. The covalent linkage of coumarin and pyrene in probe NYV extends conjugation, shifting the absorption to 500 nm and emission at 575 nm.
Comments 3: Oines 162. It is mentioned that pyrene provides the property of AIE and substantial Stokes shift. Please discuss more clearly these two properties for the hybrid molecule NYV. In other words, quantify the AIE property and mention what is the Stokes Shift of NYV. Please consider the dual emission band of this molecule (Figure 1)
Response 3: We sincerely appreciate the reviewer’s insightful comment regarding the clarification of the AIE property and Stokes shift of probe NYV. Below, we provide a detailed response and the revised content to address these points.
The AIE characteristics of probe NYV were quantitatively evaluated by measuring the fluorescence emission of probe NYV in a dilute solution (acetonitrile) and aggregated state (acetonitrile/water mixture). As shown in Figure S3, when the water fraction exceeds 60%, the fluorescence intensity of probe NYV increases significantly. This confirms the typical AIE behavior, in which the restricted intramolecular motion in the aggregated state suppresses non-radiative decay and enhances emission. The AIE-active pyrene component in probe NYV plays a key role in this phenomenon, ensuring high sensitivity even in biological or environmental matrices where aggregation is inevitable.
The Stokes shift of probe NYV was determined by analyzing the difference between the maximum absorption wavelength and the emission wavelength of probe NYV. It can be observed from the fluorescence emission spectrum (Figure 1) that under excitation at 400 nm, probe NYV presents a fluorescence emission peak at 575 nm. The maximum absorption wavelength of probe NYV is 500 nm. Therefore, the Stokes shift, emitted at 575 nm, is calculated to be 75 nm.
Comments 4: Lines 238. What does “fluorescent illumination” mean? Is it referred to UV illumination?
Response 4: Thank you for your valuable comment. The term "fluorescent illumination" indeed refers to UV illumination. To avoid any confusion, we will replace "fluorescent illumination" with "UV illumination (365 nm)" in the revised manuscript (Lines 266–267, Page 9).
Comments 5: Line 210. Is it correct the “Despite interference”? Plots in Figure 2 don’t show any interference
Response 5: Thank you for your careful review and valuable comment. We appreciate your observation regarding the statement about interference in the manuscript. Upon re-examining the data, we agree that the plots in Figure 2 primarily demonstrate the selectivity of probe NYV toward HClO over other analytes rather than explicitly showing interference effects. The interference studies are indeed detailed in Figure S2 of the Supplementary Material, where the probe’s response to HClO remains consistent even in the presence of other species. Therefore, we removed the phrase "Despite interference" and replaced it with a more accurate description: "without significant interference (Lines 225–226, Page 7).
Comments 6: Figure 3. The upper row Light stand for White light illumination?.
Response 6: Thank you for your valuable feedback and suggestions. We appreciate your attention to detail. To clarify and enhance the readability of the figure, we removed the phrase "visible light" and replaced it with a more accurate description: "white light" (Line 275, Page 9).
Comments 7: What is the site within the NYV molecule which sense HClO?
Response 7: The carbon-carbon double bond present between coumarin and pyrene serves as the recognition site of HClO in probe NYV. This finding aligns with previous studies that have demonstrated the role of carbon-carbon double bonds in various chemical structures as response sites for HClO through oxidation reactions (Chemical Science 2018, 9, 8207–8212; Sensors and Actuators B: Chemical 2018, 276, 8–12; Spectrochimica Acta Part A: Molecular and Biomolecular Spectroscopy 2024, 321, 124754; Journal of Hazardous Materials 2021, 418, 126243; Talanta 2019, 194, 308–313; Spectrochimica Acta Part A: Molecular and Biomolecular Spectroscopy 2022, 279, 121490; Spectrochimica Acta Part A: Molecular and Biomolecular Spectroscopy 2020, 229, 118001; Talanta 2021, 225, 122030; Analytical Chemistry 2021, 93, 15696–15702).
Reviewer 3 Report
Comments and Suggestions for Authors
Overall the work is well-structured. This contribution should be considered for publication after addressing the following comments:
- Rewrite and organize the abstract. some sentences should be corrected. The sentence beginning with “The probe's potential applications… could these results highlight the potential of NYV-based sensors….. also, there are some grammar and minor Formatting errors as well as lake of some numerical data like linear range , limit of detection etc. Moreover the keyword should be more specific. Write also the name of probe (NYV) before the abbreviation for the first time.
- In the experimental part Deionized water was used all over the work write the conductivity also. Additionally, modified environmental water samples were collected… write more details
- Section 2.3 clarify , Compound 1 , The reaction mixture was subjected to condensation and reflux for 14 h, what was the temperature? Please write the synthesis in detail with conditions. with progress monitored via thin layer chromatography (TLC), write condition as well as the solvent used, ratio etc.
- In section 2.4 real sample preparation, microfiltration need specification which filter what is the size were used. The resulting fluorescence intensity was detected and recorded at 460 and 575 nm… explain in detail.
- Modified figure 1 D, the analytical parameter like slope and intercept are very small, also confirm the detection limit of 45 nM.
- Some figures need error bars.
- In table 1 Recover (%) is low in Yihe water while using low concentration 30.0 (μM)
Author Response
Overall the work is well-structured. This contribution should be considered for publication after addressing the following comments:
Comments 1: Rewrite and organize the abstract. some sentences should be corrected. The sentence beginning with “The probe's potential applications… could these results highlight the potential of NYV-based sensors….. also, there are some grammar and minor Formatting errors as well as lake of some numerical data like linear range , limit of detection etc. Moreover the keyword should be more specific. Write also the name of probe (NYV) before the abbreviation for the first time.
Response 1: Thank you for your valuable comment. We have carefully revised the abstract to address the issues you raised. Below is the revised version with the necessary corrections and additions:
Revised Abstract: Hypochlorous acid (HClO) serves as a biological mediator and is widely utilized as a disinfectant in food processing and water treatment. However, excessive HClO residues in food and environmental water raise concerns due to the potential formation of carcinogenic chlorinated byproducts and disinfection byproducts (DBPs). Despite its importance, traditional methods for HClO detection often involve complex sample preparation, sophisticated instrumentation, and skilled operators. Herein, we report an aggregation-induced emission (AIE) small molecule fluorescent probe (NYV) that integrates colorimetric and ratiometric fluorescence responses for the detection of HClO. This probe exhibits high sensitivity, with a detection limit of 0.35 μM, a rapid response time of 1 min, and a wide linear range (0–142.5 μM), along with anti-interference capabilities, making it suitable for real-time monitoring. Furthermore, we have developed a portable solid-state sensor based on probe NYV for the rapid visual detection of HClO. The potential applications of this probe in real sample analysis and bioimaging experiments are demonstrated. Our findings contribute to the development of innovative fluorescent probes for HClO detection, with broad applications in food safety, environmental monitoring, and biomedical research on oxidative stress and ferroptosis (Lines 14–29, Page 1).
Revised Keywords: Small molecule fluorescent probe, Hypochlorous acid, Aggregation-induced emission, Environmental monitoring, Oxidative stress imaging, Ferroptosis imaging (Lines 30–31, Page 1).
Comments 2: In the experimental part Deionized water was used all over the work write the conductivity also. Additionally, modified environmental water samples were collected… write more details
Response 2: Thank you for your insightful comments and suggestions to improve our manuscript. In response to your comment regarding the conductivity of the deionized water used, we have added a sentence to the Materials and Chemicals subsection to describe the conductivity of the deionized water (Line 87, Page 2). Additionally, to provide more details about the environmental water samples, we have expanded the Real samples preparation subsection to include information about the collection process, storage conditions, and pre-treatment steps (Lines 89–95, Pages 2–3).
Comments 3: Section 2.3 clarify , Compound 1 , The reaction mixture was subjected to condensation and reflux for 14 h, what was the temperature? Please write the synthesis in detail with conditions. with progress monitored via thin layer chromatography (TLC), write condition as well as the solvent used, ratio etc.
Response 3: We appreciate the reviewer’s constructive feedback. The synthesis details have been revised as follows (inserted in Section 2.3 of the manuscript):
Revised Section 2.3: Compound 1 was synthesized using the existing literature (Meng, Yongbin, Fangjun, Jianbin, & Shaomin, 2022). Specifically, Compound 1 (518.24 mg, 2 mmol) and pyrene-1-carboxaldehyde (460.14 mg, 2 mmol) were dissolved in absolute ethanol (40 mL), and 100 μL of piperidine was added as a catalyst. The reaction mixture was condensed and refluxed at 80 °C for 14 h, and the process was monitored by thin-layer chromatography (TLC, Rf = 0.37) using a silica gel plate (GF254) with a mobile phase of dichloromethane (CH2Cl2). The TLC plates were visualized under UV light (365 nm). Upon completion of the reaction, the solution was subjected to suction filtration to isolate an orange solid. The crude product was purified via silica gel column chromatography using CH2Cl2 as the eluent, yielding probe NYV (289 mg, 0.61 mmol) (Lines 103–112, Page 3).
Comments 4: In section 2.4 real sample preparation, microfiltration need specification which filter what is the size were used. The resulting fluorescence intensity was detected and recorded at 460 and 575 nm… explain in detail.
Response 4: Thank you for pointing out the need for clarification. We have revised Section 2.4 to include the missing details as follows: The filter material (PES) and pore size (0.22 μm) are now explicitly stated; The fluorescence detection parameters (slit width, PMT voltage, and ratiometric calculation) are detailed to ensure reproducibility.
Revised Section 2.4: Before analysis, real samples of tap water, Luo River water, and Yi River water were subjected to microfiltration using a 0.22 μm polyethersulfone (PES) membrane filter to remove insoluble particles and ensure sample clarity. Each sample was reacted with probe NYV (10 μM) for 1 min, after which the fluorescence intensity was measured. Additionally, spike recovery testing necessitated the precise addition of HClO at concentrations of 0, 30, 60, and 90 μM. The resulting fluorescence intensity was detected and recorded at 460 and 575 nm with an excitation wavelength of 400 nm. Each experiment was repeated 3 times. During fluorescence detection, the sample was excited at 400 nm, and the emission intensifications were recorded at 420–700 nm using the Agilent Cary Eclipse 9800A fluorescence spectrophotometer. The slit widths for both excitation and emission are set at 10 nm, and the voltage of the photomultiplier tube (PMT) is maintained at 700 V to ensure consistent sensitivity. The fluorescence intensity ratio (I460/I575) was calculated to quantify the concentration of HClO, as this ratio method minimizes environmental interference and improves the reliability of detection (Lines 120–133, Page 3).
Comments 5: Modified figure 1 D, the analytical parameter like slope and intercept are very small, also confirm the detection limit of 45 nM.
Response 5: We sincerely appreciate the reviewer’s valuable feedback regarding Figure 1D. The linear correlation between the emission intensity ratio (I460/I575) of probe NYV (10 μM) and HClO concentrations (0–142.5 μM) is presented in Figure 1D. The calibration curve yielded a slope of 0.01945 μM-1 and an intercept of 0.002255472, with a correlation coefficient (R2) of 0.998. The detection limit (LOD) was calculated to be 0.35 μM using the formula LOD = 3σ/K, where σ is the standard deviation of the blank measurement (n = 15, Table S1) and K is the slope of the calibration curve. This confirms the high sensitivity of probe NYV for HClO detection.
Table S1 Standard deviation (Ï) of blank measurement.
λem = 491 nm |
|
0.283508307 |
|
0.287140562 |
|
0.285834718 |
|
0.281738455 |
|
0.284604217 |
|
0.282128619 |
|
0.282515318 |
|
0.283015688 |
|
0.287260818 |
|
0.281058761 |
|
0.286263623 |
|
0.283376401 |
|
0.287032085 |
|
0.287784619 |
|
0.285578237 |
|
σ |
0.002255472 |
Comments 6: Some figures need error bars.
Response 6: Thanks for your helpful comments and advice. We added error bars to the applicable range of pH values (Figure 2d, Line 230, Page 8).
Comments 7: In table 1 Recover (%) is low in Yihe water while using low concentration 30.0 (μM)
Response 7: We sincerely thank the reviewers for their profound observations on the relatively low recovery rate in Yihe water at a spiked concentration of 30.0 μM. To solve this problem, we conducted additional experiments to verify the robustness of the method and included detailed explanations in the revised manuscript (Table 1, Line 278, Page 9).
Round 2
Reviewer 1 Report
Comments and Suggestions for Authors
I have previously reviewed this manuscript. The authors have made substantial improvements based on the reviewers’ comments. In its current form, the manuscript is suitable for acceptance.
Author Response
Thank you for your recognition and contribution to the improvement of the manuscript.
Reviewer 2 Report
Comments and Suggestions for Authors
Thanks to authors for attending my comments and suggestion for numerals 2-7. As for comments of numeral 1 , I could not see the Jablonsky diagram authors included in their s response. In any case, the claim that this molecule exhibits an anti Kasha's rule emission is not well supported. There are few molecules able to emit efficiently from S2; in fact, many effects mimic such S2 – S0 emission. I suggest reviewing this issue more carefully. In case the emission 425 – 500 nm (Figure 1) is indeed due to S2 – S0 transitions, the effect must be clearly stated and discussed in the manuscript because of its scientific relevance including the Jablonsky diagram. Citation of literature on the same effect, and in similar molecules, would be also appreciated by readers.
Author Response
Comments 1: Thanks to authors for attending my comments and suggestion for numerals 2-7. As for comments of numeral 1 , I could not see the Jablonsky diagram authors included in their s response. In any case, the claim that this molecule exhibits an anti Kasha's rule emission is not well supported. There are few molecules able to emit efficiently from S2; in fact, many effects mimic such S2 – S0 emission. I suggest reviewing this issue more carefully. In case the emission 425 – 500 nm (Figure 1) is indeed due to S2 – S0 transitions, the effect must be clearly stated and discussed in the manuscript because of its scientific relevance including the Jablonsky diagram. Citation of literature on the same effect, and in similar molecules, would be also appreciated by readers.
Response 1: We sincerely thank the reviewer for the professional opinions. The reviewer correctly pointed out that anti Kasha's rule emissions are rare and require strong evidence. We admit that our initial manuscript did not provide sufficient discussion mechanisms or literature to support this specific claim.
Probe NYV is designed to contain a single chromophore with two different emission centers. Among them, Compound 1 is a coumarin-like fluorophore, and its fluorescence emission peak is at 478 nm. The relative rotation between coumarin and pyrene groups may affect the intramolecular energy transfer and emission processes. Additionally, the probe exhibits an AIE effect, and the fluorescent probe may aggregate. Compared with the monomer state, the aggregated state has different molecular interactions and electronic environments. The above may affect the fluorescence emission of the probe.
We also found that some literature reports similar phenomena. That is, under the action of the excitation wavelength, a single probe presents two fluorescence emission peaks (one of which has a lower intensity) (Spectrochim. Acta. A Mol. Biomol. Spectrosc. 2022, 270, 120827; Sens. Actuators B Chem. 2022, 373, 132742; Chem Commun 2022, 58, 3633-3636; Chem Commun 2021, 57, 5810-5813; Anal. Chim. Acta 2022, 1203, 339652; Anal Sci 2021, 37, 1541-1546; J Hazard Mater 2021, 413, 125332; Spectrochim. Acta. A Mol. Biomol. Spectrosc. 2023, 302, 123128; Sensors 2019, 19, 5348; ACS Omega 2018, 3, 10145-10153; Anal. Chem. 2020, 92, 11396-11404; Spectrochim. Acta. A Mol. Biomol. Spectrosc. 2019, 221, 117175; Chin. Chem. Lett. 2022, 33, 2527-2531; Anal. Chem. 2016, 88, 1455-61; Sens. Actuators B Chem. 2023, 389, 133841; Anal. Chem. 2022, 94, 4763-4769; Talanta 2019, 202, 190-197; Talanta 2024, 267, 125157; Sens. Actuators B Chem. 2022, 369, 132352; Talanta 2021, 223, 121720; Sens. Actuators B Chem. 2024, 407, 135453; Journal of Hazardous Materials 2025, 487, 137151; Sens. Actuators B Chem. 2023, 392, 134064; Chem. Sci. 2022, 13 (18), 5363-5373; Spectrochim. Acta. A Mol. Biomol. Spectrosc. 2022, 283, 121736; Anal. Chem. 2021, 93, 5194-5200; Anal. Chim. Acta 2021, 1187, 339159; Sens. Actuators B Chem. 2022, 361, 131751).
We fully agree with the reviewer's point of view and understand the reviewer's concerns. Therefore, the relevant content awaits further detailed research, and the research results will be published separately.
Reviewer 3 Report
Comments and Suggestions for Authors
The author has addressed all the comments, so the paper can now be accepted
Author Response

(The authors gave the same response as above.)

Round 3
Reviewer 2 Report
Comments and Suggestions for Authors
Thanks to authors for the answer to my suggetions. I suggest to include in the manuscript a couple of sentences regarding the issue of anomalous emission. Certainly the issue awaits further detailed research, but readers would appreciate that author bring to the attention such a issue.
Author Response
Comments 1: Thanks to authors for the answer to my suggetions. I suggest to include in the manuscript a couple of sentences regarding the issue of anomalous emission. Certainly the issue awaits further detailed research, but readers would appreciate that author bring to the attention such a issue.
Response 1: Thanks for your comments and advice. We agree that emphasizing the abnormal double-emission behavior of probe NYV will enhance the scientific rigor of the paper and alert readers to this interesting phenomenon. As suggested, we have added a brief statement in the Results and Discussion section (Section 3.1) to acknowledge this observation and emphasize its significance for future research (Lines 205–215, Page 7). And it is pointed out that similar phenomena have already been reported in the literature (References 17, 22, 31, 32, 33, 40, 44). The revised text highlights this anomaly and clarifies that the origin of its mechanism requires further research, which is consistent with the reviewers' suggestions.
Revised Section 3.1: To investigate the recognition capability of probe NYV towards hypochlorous acid (HClO), we comprehensively analyzed its UV-Vis and fluorescence spectra. Figure 1a and 1b illustrate the spectral changes of probe NYV before and after HClO recognition. The probe shows a significant decrease in absorption at 500 nm and a reduction in fluorescence intensity at 575 nm, while intensity increases at 460 nm, indicating a ratiometric change that enhances HClO detection accuracy. The emission spectra of the free probe exhibit the unusual characteristics of two distinct emission peaks. This phenomenon may be related to the molecular structure of the probe, which contains a chromophore with two different emission centers. The relative rotation between coumarin and pyrene groups may affect the intramolecular energy transfer and emission processes. Moreover, the probe exhibits an AIE effect, which may cause the fluorescent probe to aggregate, potentially affecting intramolecular energy transfer and emission processes. Although the exact mechanism of this abnormal emission requires further research, it is worth noting that similar phenomena have already been reported in the literature (Gong et al, 2023; Hu et al, 2022; Li, Liu, Gao, Sheng, & Zhu, 2022; Liu et al, 2023; Liu, Ma, & Lin, 2022; Sreelaya, Drisya, & Chakkumkumarath, 2025; Wang et al, 2019). As the concentration of HClO increases, the color of the probe solution changes from yellow to colorless (Figure 1a), and the fluorescence changes from orange-yellow to blue (Figure 1b). This dual change confirms the effective recognition capability of probe NYV for HClO and provides a simple detection method. A good linear relationship exists between the fluorescence intensity of probe NYV and the concentration of HClO, particularly from 0−142.5 μM, with a detection limit of 0.35 μM (Figure 1c and 1d, Table S1). These results show that probe NYV is highly sensitive and accurately detects HClO at low concentrations, making it an efficient and intuitive tool for HClO recognition with significant research value and broad application potential.